# Dynamics of Dissolved Carbon in Subalpine Forest Streams

Jianfeng Hou, Fei Li, Zhihui Wang, Xuqing Li , Rui Cao and Wanqin Yang *

School of Life Science, Taizhou University, Taizhou 318000, China; mehoujianfeng@163.com (J.H.);
flissrs@yeah.net (F.L.); 17816890031@163.com (Z.W.); xuqing.li@outlook.com (X.L.); cr_leshan@163.com (R.C.)
* Correspondence: scyangwq@163.com

**Abstract:** Dissolved carbon (DC) in forest streams plays a crucial role in maintaining the structure and productivity of adjoining aquatic ecosystems as well as informing biogeochemical links between mountain forests and adjoining rivers. Nevertheless, the functions of forest stream DC dynamics are rarely incorporated into river management. To better understand the biogeochemical links between subalpine forests and adjoining streams, the seasonal dynamics of DC in 15 representative forest streams were investigated in a geographically fragile subalpine-gorge catchment in the upper reaches of the Yangtze River. Depending on stream characteristics and critical periods, the DC stocks in the streams ranged from 0.22 to 2.35 mg m$^{-2}$ for total DC, from 0.10 to 1.66 mg m$^{-2}$ for dissolved inorganic carbon (DIC), and from 0.12 to 1.27 mg m$^{-2}$ for dissolved organic carbon (DOC). Moreover, the annual stocks of DC, DIC, and DOC were 1.01, 0.56, and 0.45 mg m$^{-2}$, respectively. Correspondingly, the averaged export rates for DC, DIC, and DOC from the forest streams ranged from 0.27 to 1.98 mg s$^{-1}$, from 0.24 to 1.48 mg s$^{-1}$, and from 0.18 to 0.90 mg s$^{-1}$, respectively, in the subalpine forest catchment. The annual export rates of total DC, DIC, and DOC were 1.06, 0.75, and 0.31 mg C s$^{-1}$, respectively. In particular, the highest rates of export were 4.67, 3.53, and 1.34 mg s$^{-1}$ for DC, DIC, and DOC, respectively, in the snowmelt period. The average ratios of DOC to DIC stock in the export water ranged from 0.23–2.41 for the 15 streams, and the average value was 0.85 during this one-year investigation. In addition, the maximum and minimum values of the DC stocks, their exports, and the DIC:DOC ratio were consistently observed during the snowmelt season and the late growing period. In summary, precipitation, temperature, water discharge rate, and sediment depth regulated the stocks and export rates of DC and its components. In general, forest streams are important links between the carbon biogeochemical cycle of subalpine forests and adjoining streams.

**Keywords:** carbon budget; dissolved carbon; dissolved inorganic carbon; dissolved organic carbon; forest stream

## 1. Introduction

Forest streams are similar to peripheral neural networks or capillaries in mountainous areas that link the terrestrial–aquatic biogeochemical cycles. On the one hand, the surface waters of these streams are conduits through which terrestrial carbon can be returned to the atmosphere by emission or can be exported to the ocean, and they play an irreplaceable role in the global carbon cycle [1,2]. On the other hand, mountain forests and riparian vegetation are major sources of organic carbon to adjoining aquatic ecosystems [3,4], and carbon inputs to streams play a crucial role in maintaining the structure and productivity of aquatic ecosystems [5]. In particular, dissolved carbon (DC) is an important form of dissolved carbon import from the forest and riparian vegetation [6]. Therefore, a thorough investigation of the dynamics of carbon in forest stream waters is helpful for understanding the biogeochemical links between forests and aquatic ecosystems and for providing a key scientific basis for managing forest catchments and predicting the global carbon cycle.

DC includes dissolved organic and inorganic C (DOC and DIC, respectively), which influence the surface water chemistry [6,7], e.g., in metal bioavailability and mobility [8], nutrient cycling [9], pH buffering, and ionic balance [10]. From a theoretical perspective,

forests and riparian zones import DC into adjacent streams and rivers through plant residues (woody and nonwoody debris) [2,11,12], surface runoff [13,14], and percolating water [11,15]. These processes vary greatly with climatic factors (e.g., precipitation and temperature), vegetation stages, vegetation types, and characteristics of streams [16–19]. However, the effects of these factors on the dynamics of DC storage, composition, and export in stream water have not been reported; therefore, it is difficult to understand the actual roles of forest streams in upland forests and adjoining aquatic ecosystems, especially in geographically fragile mountain regions. This subject needs to be fully investigated.

The stock of DC (DOC and DIC) in streams embedded in forest catchments is mainly affected by three factors: climatic factors (temperature, precipitation, etc.), stream traits (length, width, discharge velocity, sediment, etc.), and vegetation characteristics (forest type and plant growth rhythm) [16,17,20–22]. First, precipitation and snowmelt are the two main regulators of DC stocks [23–25]. For example, the increase in runoff combined with the increase in percolating water caused by precipitation is consistently accompanied by an increase in DOC and DIC concentrations, which can also increase their exports [14]. Additionally, riverbank soil scouring by the unsteady water discharge caused by seasonal precipitation can increase both the DOC and DIC stocks [26]. Second, soil temperature and humidity can affect the fraction and solubility of soil organic carbon, which can also affect the DC stocks [27]. Third, changes in environmental factors during critical periods can influence plant species and their growth rhythms, which can then influence the DC stocks by regulating litter inputs to neighboring streams [11,28] and by their decomposition rates in the streams [12]. Furthermore, stream characteristics, such as their area, sediment depth, and discharge velocity, are the main regulators of carbon dynamics in subalpine forest streams [11,12]. However, some investigations have found that DOC stocks are somewhat correlated with stream characteristics [29,30], but the relationships between DC stocks in forest streams and climatic factors and stream characteristics remain unclear. Moreover, there is a lack of information on stream DC chemistry in most headwaters of mountainous subalpine forest regions.

As a major source of OC in forested catchments, the OC initially moves from soils (terrestrially derived C) to inland waters. In aquatic ecosystems, partial OC is released into the atmosphere through outgassing (about 40% of the total terrestrially derived C) or buried in the sediment (about 20%), and the remainder (about 40%) is eventually delivered to the coastal ocean [31,32]. Aquatic OC export from forest streams is determined by the interactive effects of their stocks, climate (temperature, precipitation, etc.), vegetation types and stages, hydrologic characteristics (stream size, discharge velocity, etc.), and nitrogen and sulfur deposition [31–33]. First, the magnitude of DC input to the streams determines the size of their source of an adjoining river [28,34]. Second, the exports of DC from streams are altered by a variety of processes, including primary production, microbial breakdown, sorption to particles, and photodegradation within aquatic systems, which are mainly regulated by climatic factors, such as precipitation and temperature [35]. For example, the quantity of DOC and DIC exported from streams consistently increases with rainfall during the growing seasons, while it has lower levels during snow-covered periods [36,37]. Third, the exports of DC also vary greatly with landscape pattern changes, and spatial variations in DC export are often caused by the existing catchment vegetation and soil properties [38]. For example, stream DC export often increases during spring floods in forested catchments, owing to the rising water tables that flush out DC-rich water that was previously stored in the soil [39]. Fourth, DC export from forest streams consistently depends upon stream characteristics (length, width, discharge velocity, etc.) and often varies with downstream distance [40]. Finally, the nitrogen and sulfur deposition can accelerate the soil DOC leaching by increasing the soil DOC production and controlling the soil sorption ability [32]. Consequently, there are large variations in DC export on spatial [41–43] and temporal scales [19,44]. Therefore, thorough investigations into the DC export from forest streams can facilitate a better understanding of the biogeochemical links between mountain forests and riparian zones and aquatic ecosystems.

Subalpine forest streams are considered one of the most important biogeochemical carriers between mountain–forest ecosystems and adjoining aquatic ecosystems [45,46]. Distinct temperature and heavy precipitation fluctuation patterns are regular for these regions [46–49]; thus, dynamic changes of these factors can contribute differently to the DC dynamics during different seasons. Therefore, a thorough investigation of the dynamics of the DOC and DIC budget of these streams during critical periods is key not only for revealing the carbon biogeochemical linkages between subalpine forest and aquatic ecosystems but also for providing a scientific basis for managing the upper reaches of the Yangtze River. This study is expected to further elucidate the DC stocks, DOC/DIC ratios, and their exports as well as the confluence characteristics of subalpine forest streams by monitoring the dynamic changes in DC in subalpine forest streams. We hypothesized that (1) the DOC and DIC stocks and exports might have different dynamic patterns during critical periods, such as the growing season, snow-covered season, and snowmelt season; (2) the peaks and valleys values of the indices mentioned above would appear during the later growing and snowmelt periods, respectively; and (3) the DOC and DIC inputs to the waters would be higher than the exports from them for this research region.

## 2. Materials and Methods

### 2.1. Site Description

This study was conducted at the Long-term Research Station of the Subalpine Forest Ecosystem in the Bipenggou Valley (31°14′~31°19′ N, 102°53′~102°57′ E, 2458~4619 m.a.s.l.), Li County, Southwest China, which is in a subalpine gorge area situated between the Tibetan Plateau and the Sichuan Basin. It has frequent geological breaks, clear seasonal snow cover (the maximum snow depth is approximately 35 cm), frequent freeze–thaw cycles, and the soil is dark brown soil (GB/T 17296-2009) [11,50]. The highest and lowest temperatures are 23 °C (July) and −18 °C (January), respectively. The frozen season lasts from November to April and begins in late April [47]. The detailed characteristics of this catchment are shown in Table 1.

**Table 1.** Catchment characteristics (MAP, MAT, Soil C, N, P) of the investigated subalpine forest regions.

| MAP (mm) | MAT (°C) | C (g kg$^{-1}$) | N (g kg$^{-1}$) | P (g kg$^{-1}$) | Main Plant |
|---|---|---|---|---|---|
| 850.00 | 3.00 | 126.00 | 5.80 | 1.20 | Minjiang fir (*Abies faxoniana* Rehder & E.H.Wilson), Larch (*Larix mastersiana* Rehder & E.H.Wilson), Cypress (*Sabina saltuaria* Rehder & E.H.Wilson), Shrubs of azaleas (*Rhododendron* spp.), Willow (*Salix* spp.), Barberry (*Berberis argentina* C.K.Schneid), Ferns (*Cystopteris montana* (Lam.) Bernh. ex Desv) |

MAP: mean annual precipitation; MAT: mean annual temperature; C, N, P values reported were concentrations in the surface soil (0–5 cm of the organic horizon without litter layer).

### 2.2. Experimental Design

Based on previous investigations, 15 permanent streams representative of the forest catchment were selected at elevations of 3600~3700 m in this typical subalpine forest catchment (detailed characteristics of these streams are shown in [11]). The total investigated area was 4.3 km². We chose three sampling sites at the upper, middle, and end of every stream for collecting water samples, and measured the length, width, sediment depth, and discharge velocity (stream characteristics) for every stream at each sampling time. We measured the length, width, and width by a flexible ruler for every stream at every sampling site. The length was measured from the source to the estuary along their banks, and the width was measured horizontally from one bank to another bank. The stream depth was measured vertically. All these indices were measured three times, and the average was taken. We measured the discharge velocity every 30 min using a flowmeter (Martin Marten Z30, Current-meter, Barcelona, Spain), recorded the temperature every 2 h using a

button thermometer (iButton DS1923-F5, Maxim/Dallas Semiconductor, Sunnyvale, CA, USA), and measured the precipitation for real-time monitoring using a rainfall monitor (ZXCAWS600, Weather, Beijing, China) at every site.

### 2.3. Sample Collection and Treatment

We chose three sampling sites for collecting water samples at the sites mentioned above. Water samples were collected randomly from the source, middle, and end of the stream. To avoid introducing floating objects to the water surface, the sediment at the bottom of the stream was not stirred during water sampling. Then, the water samples were transported in precleaned polyethene bottles and stored at 4 °C in the dark until analysis for less than one week. The samples were passed through 0.45 μm filters, and the filtration was used to determine the concentrations of dissolved carbon and dissolved inorganic carbon using a TOC analyzer (multi N/C 2100, Analytik Jena, Thüringen, Germany).

Based on the phenological changes, seasonal precipitation, and temperature dynamics of the study area [48], we divided one year into five different periods—the snowmelt season (SMS: April to May), early growing season (EGS: May to June), growing season (GS: July to August), late growing season (LGS: September to October), and seasonal snow cover (SSC: November to April of the following year). Water was sampled nine times during the growing periods and four times during the nongrowing periods. Specifically, water samples were collected during the LGS at approximately 15 d intervals. Therefore, the DC stock and concentration at each period for the 15 streams were treated as 15 replicates during the data analysis.

### 2.4. Analytical Methods and Calculations

Each stream was approximated as a cube, so the carbon stock $S_C$ (mg m$^{-2}$) was estimated to be

$$S_C = (C_C \times V_S/S_s) \times 1000 \tag{1}$$

where $C_C$ is the carbon concentration, mg L$^{-1}$; $V_S$ is the stream cube, m$^3$ (stream length $\times$ width $\times$ water depth (m)), and Ss is the stream area, m$^2$.

The export rate $K_e$ (mg s$^{-1}$) was calculated as well:

$$K_e = (C_C \times Q_e) \times 1000 \tag{2}$$

where $C_C$ is the carbon concentration (mg L$^{-1}$) and $Q_e$ is the discharge rate of export carbon in water (m$^3$ s$^{-1}$).

The total export of the subalpine forest streams was estimated to be the sum of the carbon export from each stream. The amount of carbon export per unit area of streams $F_s$ (mg m$^{-2}$ s$^{-1}$) was estimated to be

$$F_s = K_t/S_s \tag{3}$$

where $K_t$ (mg s$^{-1}$) is the total export rate of the subalpine forest streams and $S_s$ (m$^2$) is the stream area.

The ratios of DC from source water to export water R were calculated as well:

$$R = S_{source}/S_{end} \tag{4}$$

where $S_{source}$ and $S_{end}$ are the carbon stock (mg m$^{-2}$) in the stream source and end, respectively.

Linear mixed-effect models were used to analyze the relationship of the DC stock and export with environmental factors (temperature, precipitation) and stream characteristics in these streams for different sampling periods. First, the normality of the residuals, homoscedasticity of the errors, and independence of the errors of our data were tested to determine whether they met the assumptions for analyses. Second, the sampling period was treated as a fixed effect, and the DC stock and export of 15 streams during every period as repetitions; then, we conducted repeated measures by Analysis of Variance (ANOVA) and Least Significant Difference (LSD) to examine their variability among different critical

periods. Third, to better illustrate the correlations of the relative indices of the DC stock and export with the explanatory variables (precipitation, temperature, sediment depth, discharge rate, litter carbon input), these variables were treated as fixed factors, and the stream was included as a random factor. We used linear and quadratic models to fit the relationships of these indices of DC with the changes in various explanatory variables. All analyses were conducted in R using the $LME_4$ package [51].

## 3. Results

### 3.1. The Stock for Dissolved Carbon in Forest Streams

The DC stock in the 15 subalpine forest streams ranged from 0.22 to 2.35 mg m$^{-2}$, the DIC ranged from 0.10 to 1.66 mg m$^{-2}$, and the DOC ranged from 0.12 to 1.66 mg m$^{-2}$ (Figure 1). Their average values were 1.01, 0.56, and 0.45 mg m$^{-2}$, respectively. In general, the DC, DIC, and DOC stocks in the streams showed a unimodal pattern. The value (0.99 mg m$^{-2}$) for the total DC stock began to increase gradually from the LGS and then reached its maximum value (1.47 mg m$^{-2}$) during the SMS. Then, it decreased gradually from the SMS to the LGS. The DIC and DOC also showed the same patterns as the total DC stock; 0.84 and 0.64 mg m$^{-2}$ were the peak values in SMS, and 0.53 and 0.37 mg m$^{-2}$ were their minimum values in LGS, respectively.

In contrast, according to the stream characteristics, the stock of DC showed different patterns in the 15 streams. The DC stock average ranged from 0.52 to 1.67 mg m$^{-2}$ (Figure 2A), the DIC ranged from 0.29 to 0.88 mg m$^{-2}$ (Figure 2B), and the DOC ranged from 0.23 to 0.72 mg m$^{-2}$ (Figure 2C).

### 3.2. The Export of Dissolved Carbon in Forest Streams

The average export of DC from streams ranged from 0.27–1.98 mg s$^{-1}$ for DC, 0.24–1.48 mg s$^{-1}$ for DIC, and 0.18–0.90 mg s$^{-1}$ for DOC, and their average values were 1.06, 0.75, and 0.31 mg s$^{-1}$ during different key periods, respectively (Figure 3). The highest rates of export reached 4.67, 3.53, and 1.34 mg s$^{-1}$ for DC, DIC, and DOC, respectively. In general, the dynamics of the DC export to the streams showed unimodal patterns, which showed a similar pattern to that of the total DC stocks. The minimum values all also appeared during the LGS, then began to increase gradually, and reached their maximum values (1.98, 1.48, 0.90 mg s$^{-1}$) during the SMS. Then, all of them began to decrease gradually from the SMS to the GS during our investigation periods.

The export rates of DC, DIC, and DOC for the 15 streams were 0.01–4.67, 0.01–3.33, and 0.01–2.56 mg s$^{-1}$, respectively (Figure 4). The averages were 1.06, 0.75, and 0.51 mg s$^{-1}$ for DC, DIC, and DOC, respectively.

The average ratios of DOC to DIC stock in the export water of these subalpine forest streams ranged from 0.23–2.41 for the 15 streams, and the average value was 0.85 during this one-year investigation (Figure 5). In general, the ratios showed two peaks, which appeared during the LGS (0.89) and EGS (0.88), and the lowest point was observed during the SMS (0.67) with different key periods. These values were all averaged per period.

### 3.3. The Ratios of DC from Source Water to Export Water in Forest Streams

The average ratios of the source and end stocks of total DC, DIC, and DOC in the forest streams were in the ranges of 0.99–1.07, 0.98–1.07, and 0.96–16.38, with averages of 1.03, 1.03, and 1.12, respectively (Figure 6), during different key periods.

In general, similar to the trends in the DC stock, all showed bimodal distributions, and there were no significant differences between them. In addition, these ratios showed different patterns in the 15 streams (Figure 7); the averaged values ranged from 0.93 to 1.22 for DC, 0.93 to 1.22 for DIC, and 0.89 to 1.35 for DOC for 15 streams in this one-year investigation. With the exception of some streams that had ratios >1, the ratios of most of the streams were <1.

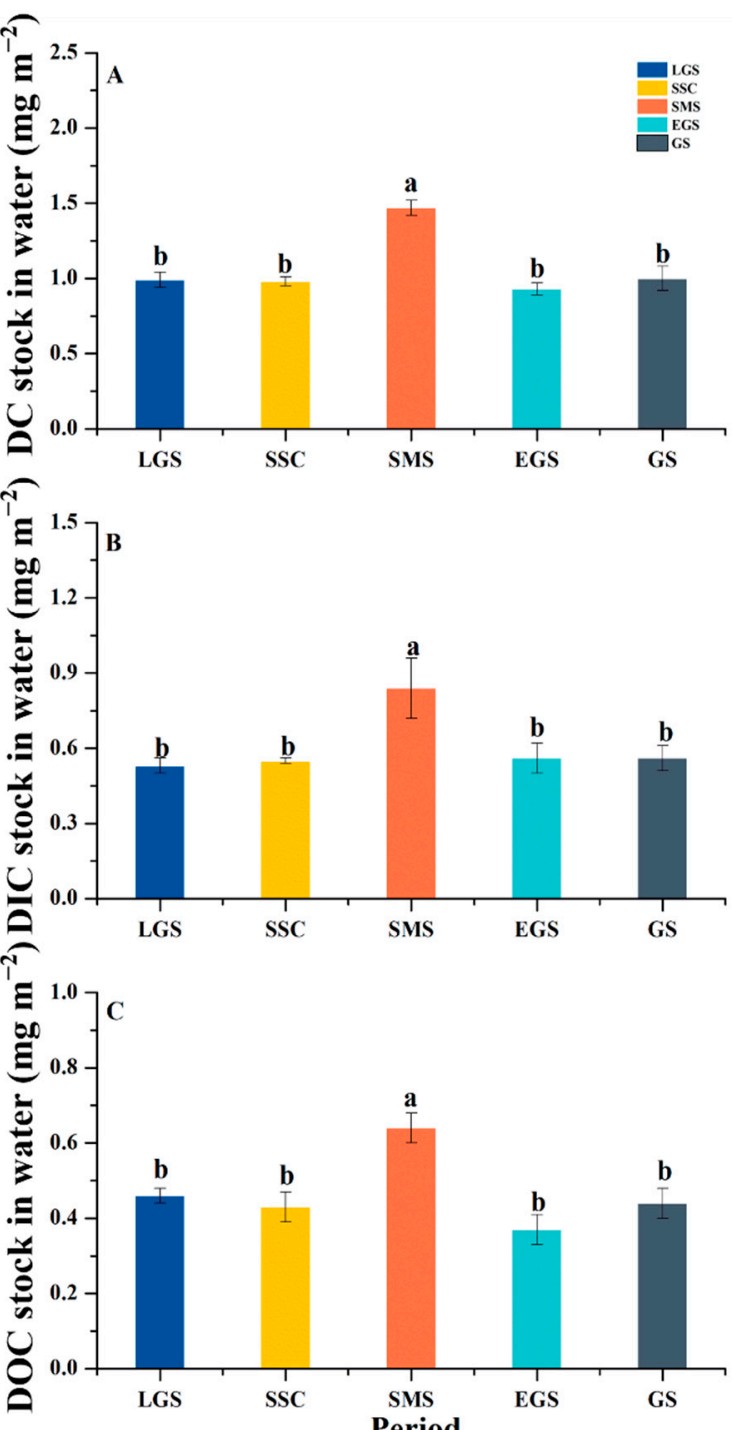

**Figure 1.** Dynamics of the dissolved carbon stock (DC), dissolved inorganic carbon stock (DIC), and dissolved organic carbon stock (DOC) in the subalpine forest streams in the upper reaches of the Yangtze River. Dissolved carbon stock (DC, (**A**)); dissolved inorganic carbon stock (DIC, (**B**)); dissolved organic carbon stock (DOC, (**C**)). LGS, SSC, SMS, EGS, and GS indicate the sampling periods, i.e., the late growing season (LGS: September to October), seasonal snow cover (SSC: November to April of the following year), snowmelt season (SMS: April to May), early growing season (EGS: May to June), and growing season (GS: July to August). The values of the vertical coordinates are the average values from a total of 15 streams in this period, and the error bars are the standard deviations of 15 streams during every period. Different lowercase letters indicate significant differences among the different periods ($p < 0.05$), while the same letter indicates no significant difference among each.

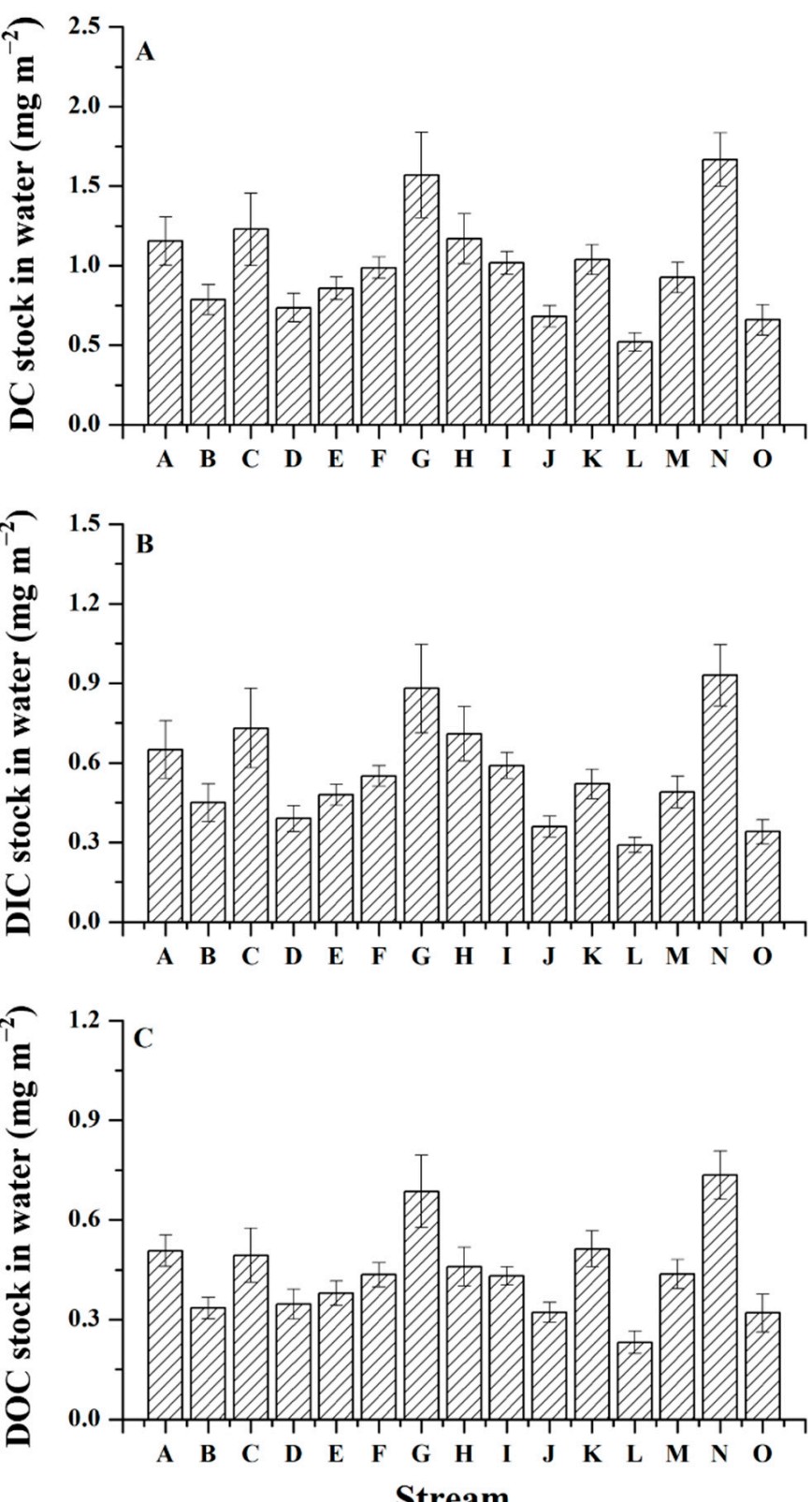

**Figure 2.** Dynamics of the dissolved carbon stock (DC), dissolved inorganic carbon stock (DIC), and dissolved organic carbon stock (DOC) in the subalpine forest streams in the upper reaches of the Yangtze River. Dissolved carbon stock (DC, (**A**)); dissolved inorganic carbon stock (DIC, (**B**)); dissolved organic carbon stock (DOC, (**C**)). Each bar is the average of 13 sampling times for each forest stream. A–O are the sampled streams in the study.

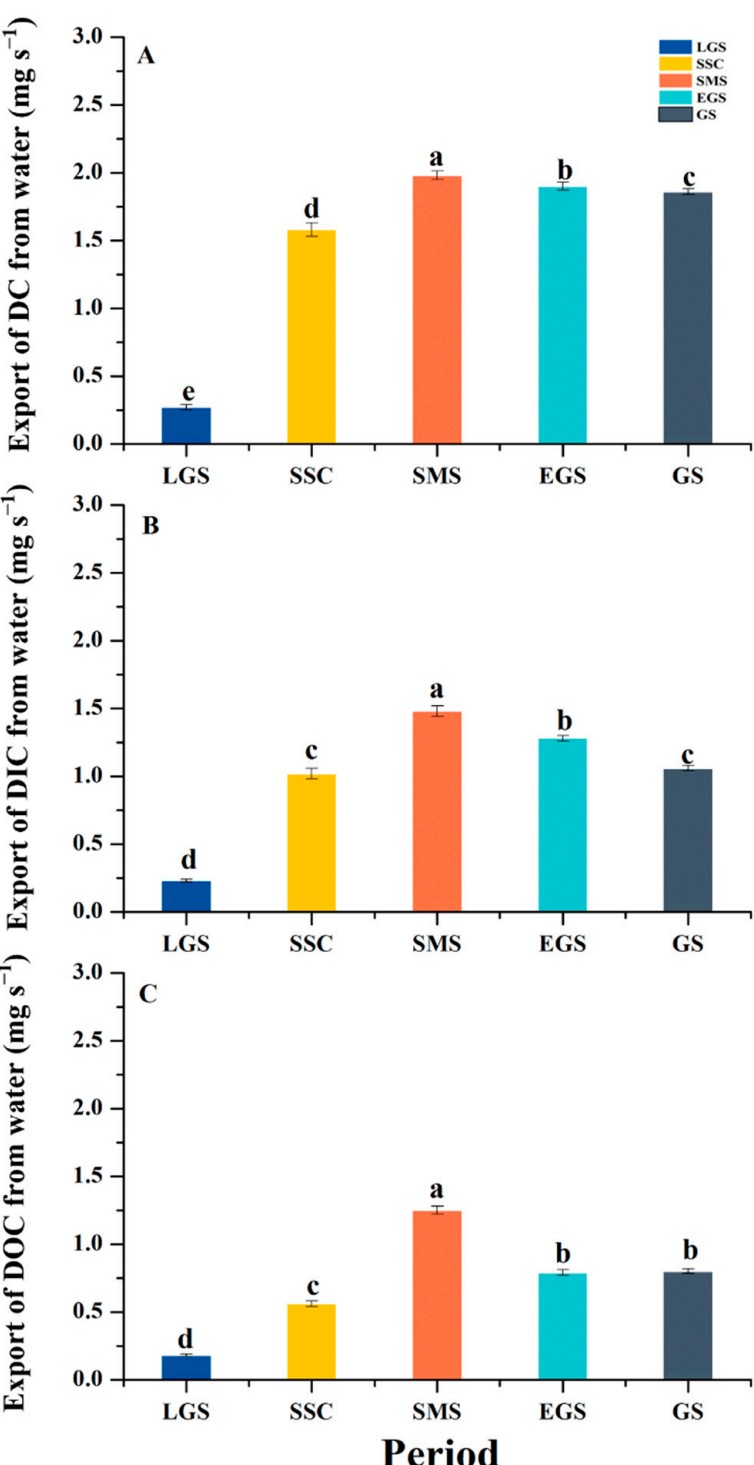

**Figure 3.** Dynamics of dissolved carbon (DC) export from the subalpine forest streams in the upper reaches of the Yangtze River. Total carbon stock (DC, (**A**)); total inorganic carbon stock (DIC, (**B**)); total organic carbon stock (DOC, (**C**)). LGS, SSC, SMS, EGS, and GS indicate the sampling periods, i.e., the late growing season (LGS: September to October), seasonal snow cover (SSC: November to April of the following year), snowmelt season (SMS: April to May), early growing season (EGS: May to June), and growing season (GS: July to August). The values of the vertical coordinates are the averaged values for 15 streams in this period, and the error bars are the stand deviations of 15 streams during every period. Different lowercase letters indicate significant differences among different periods ($p < 0.05$), while the same letter indicates no significant difference among each.

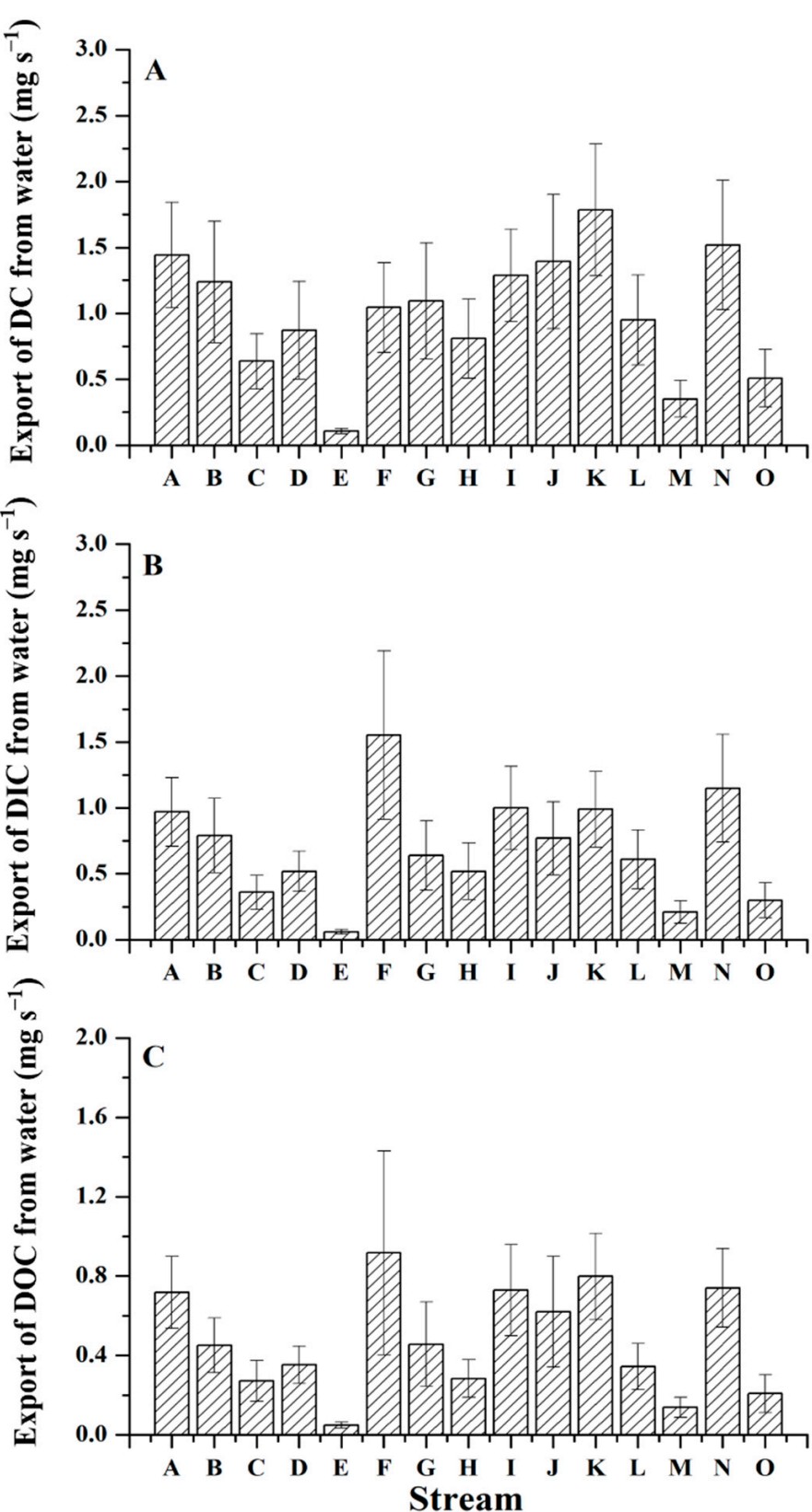

**Figure 4.** Dynamics of dissolved carbon (DC) export from the subalpine forest streams in the upper reaches of the Yangtze River. Total dissolved carbon stock (DC, (**A**)); dissolved inorganic carbon stock (DIC, (**B**)); dissolved organic carbon stock (DOC, (**C**)). Each bar is the average of 13 sampling times for each forest stream. A–O are the sampled streams in the study.

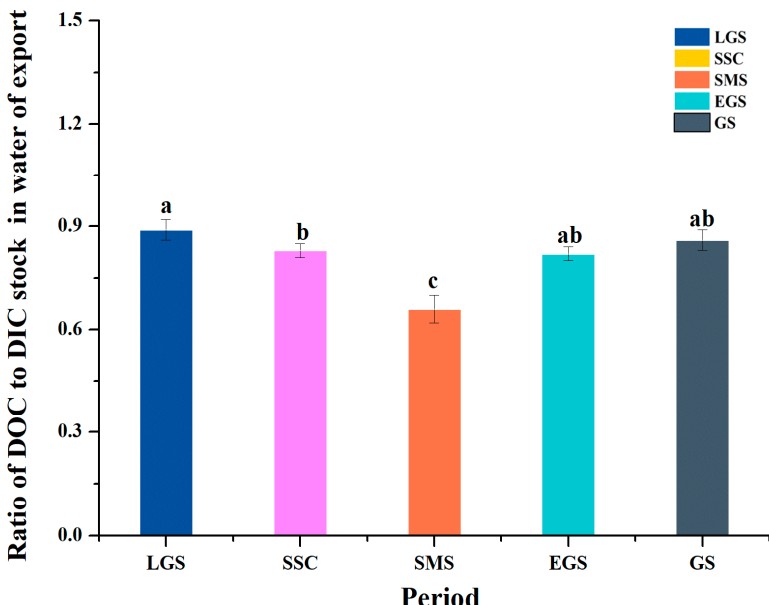

**Figure 5.** The ratios of dissolved organic carbon (DOC) to dissolved inorganic carbon (DIC) stocks in the export water of the subalpine forest streams in the upper reaches of the Yangtze River from 11 July 2015, to 2 August 2016. LGS, SSC, SMS, EGS, and GS indicate the sampling periods, i.e., the late growing season (LGS: September to October), seasonal snow cover (SSC: November to April of the following year), snowmelt season (SMS: April to May), early growing season (EGS: May to June), and growing season (GS: July to August), respectively. The values of the vertical coordinates are the averaged values for 15 streams in this period, and the error bars are the standard deviations of 15 streams during every period. Different lowercase letters indicate significant differences among different periods ($p < 0.05$), while the same letter indicates no significant difference among each.

### 3.4. Relationships of the Indices for DC, DIC, and DOC with Relative Variables

In general, the DC, DIC, and DOC stocks were all significantly and positively correlated with precipitation and discharge rate, respectively. However, the stocks of DC, DIC, and DOC were slightly and negatively correlated with temperature, sediment depth, and litter carbon input (Table 2).

**Table 2.** Relationships between dissolved carbon (DC), dissolved inorganic carbon (DIC), and dissolved organic carbon (DOC) stocks and investigated stream characteristics in the subalpine forest catchment.

|  | Factors | d.f. | $p$ | $r^2$ | Correlation |
|---|---|---|---|---|---|
| DC | Precipitation | 73 | <0.01 ** | 0.36 | + |
|  | Temperature | 73 | 0.08 | 0.02 | − |
|  | Sediment depth | 73 | 0.23 | 0.01 | − |
|  | Discharge rate | 73 | 0.04 * | 0.07 | + |
|  | Litter carbon input | 73 | 0.10 | 0.01 | − |
| DIC | Precipitation | 73 | <0.01 ** | 0.46 | + |
|  | Temperature | 73 | 0.02 * | 0.03 | − |
|  | Sediment depth | 73 | 0.10 | 0.01 | − |
|  | Discharge rate | 73 | 0.04 * | 0.05 | + |
|  | Litter carbon input | 73 | 0.46 | 0.02 | − |
| DOC | Precipitation | 73 | 0.01 ** | 0.47 | + |
|  | Temperature | 73 | 0.47 | 0.01 | − |
|  | Sediment depth | 73 | 0.32 | 0.01 | − |
|  | Discharge rate | 73 | 0.02 * | 0.08 | + |
|  | Litter carbon input | 73 | 0.97 | 0.01 | − |

* $p < 0.05$; ** $p < 0.01$; d.f.: degrees of freedom; +: positive relationship; −: negative relationship.

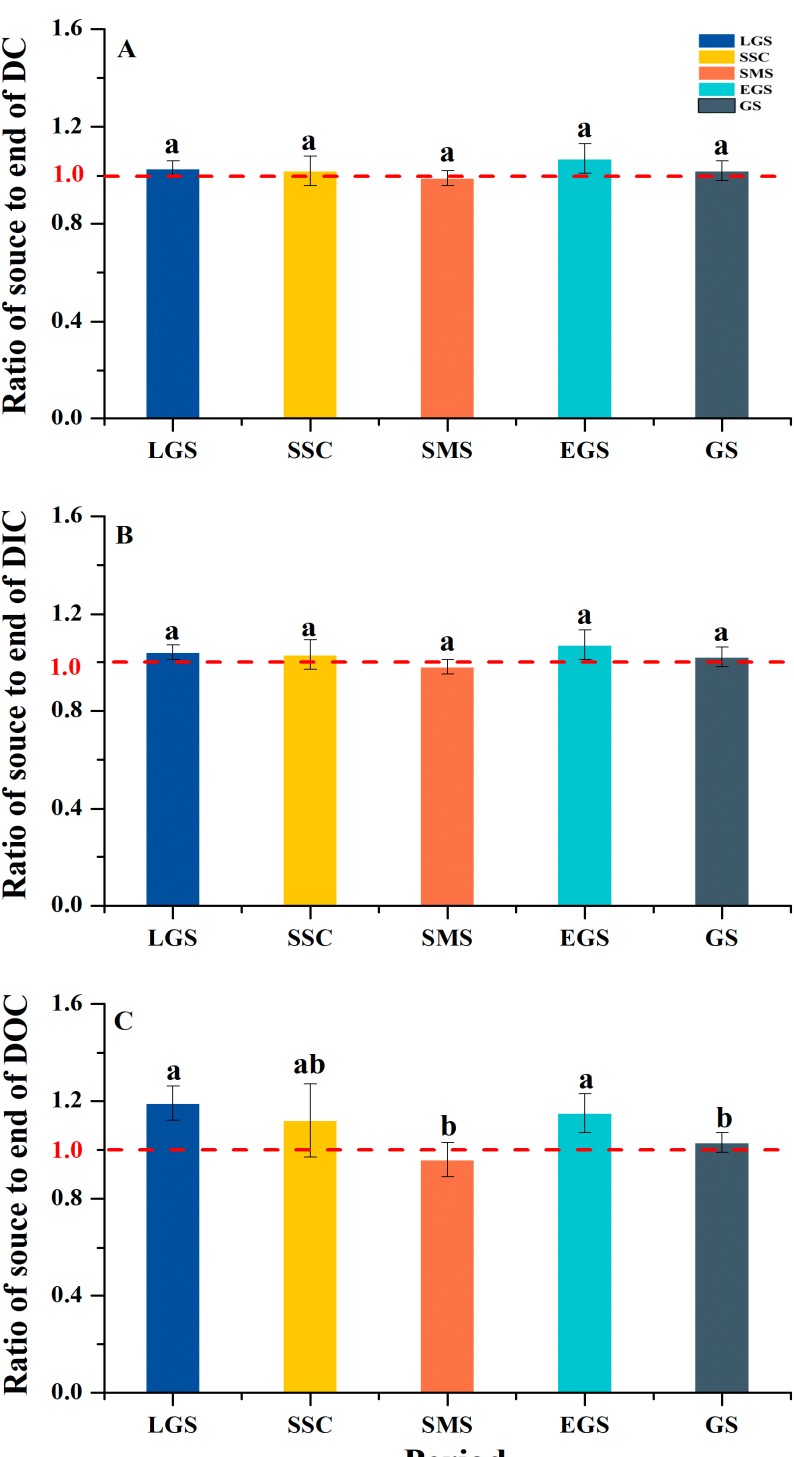

**Figure 6.** The ratios of dissolved carbon (DC) stocks from source waters to export waters in the subalpine forest streams in the upper reaches of the Yangtze River from 11 July 2015, to 2 August 2016. Total dissolved carbon stock (DC, (**A**)); dissolved inorganic carbon stock (DIC, (**B**)); dissolved organic carbon stock (DOC, (**C**)). LGS, SSC, SMS, EGS, and GS indicate the sampling periods, i.e., the late growing season (LGS: September to October), seasonal snow cover (SSC: November to April of the following year), snowmelt season (SMS: April to May), early growing season (EGS: May to June), and growing season (GS: July to August), respectively. The values of the vertical coordinates are the averaged values for 15 streams in this period, the error bars are the standard deviations of 15 streams during every period. Different lowercase letters indicate significant differences among different periods ($p < 0.05$), while the same letter indicates no significant difference among each.

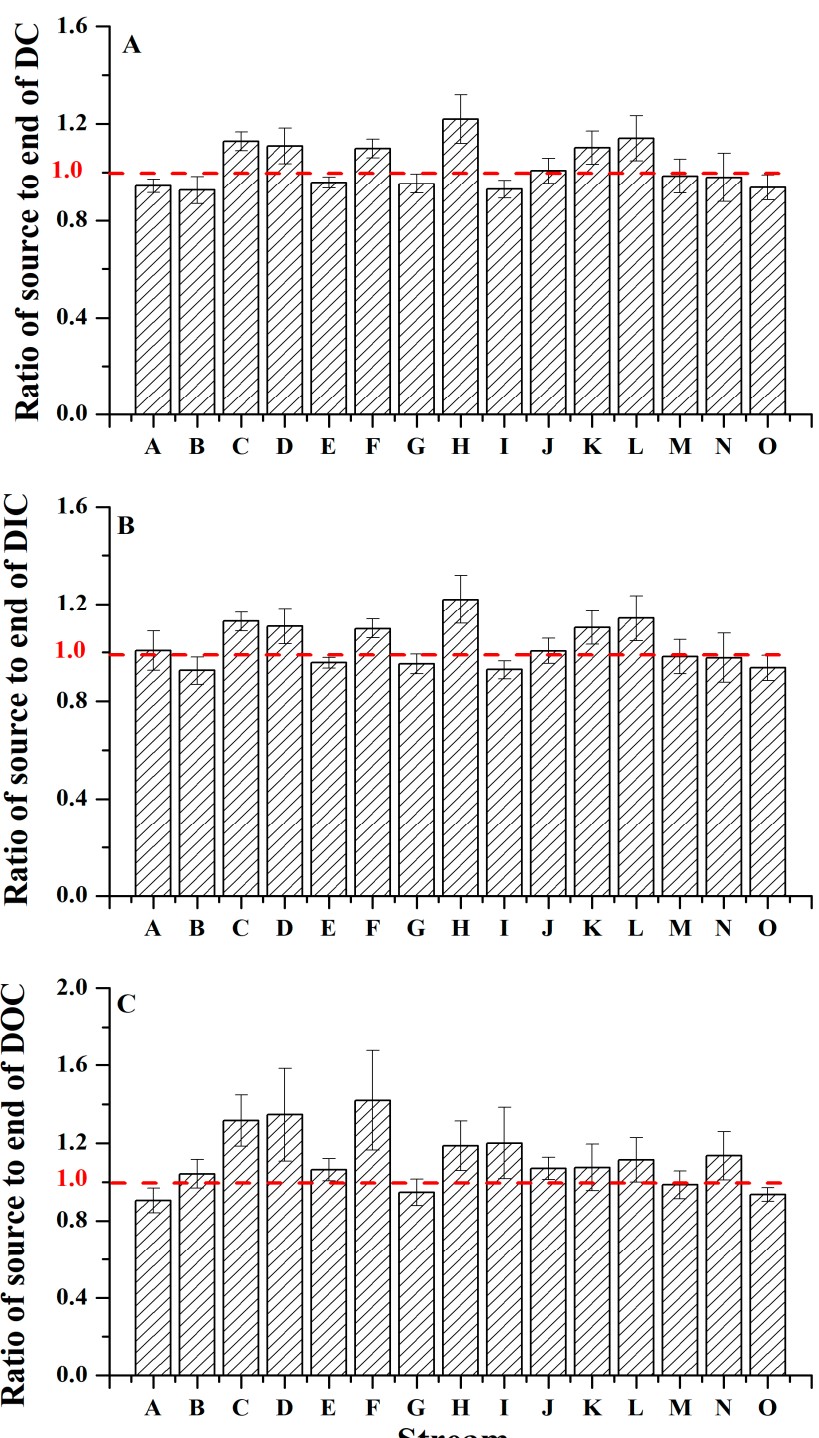

**Figure 7.** The ratios of dissolved carbon (DC) stocks from source waters to export waters in the subalpine forest streams in the upper reaches of the Yangtze River from 11 July 2015, to 2 August 2016. Total dissolved carbon stock (DC, (**A**)); dissolved inorganic carbon stock (DIC, (**B**)); dissolved organic carbon stock (DOC, (**C**)). Each bar is the average of 13 sampling times for each forest stream, and the error bars are the standard deviations of 15 streams during every period. A–O are the sampled streams in the study.

In addition, the DC, DIC, and DOC export from water were all negatively correlated with the temperature, sediment depth, and litter C input significantly. However, they were all slightly and positively correlated with precipitation, but the correlations were not significant (Table 3).

**Table 3.** Relationships between dissolved carbon (DC), dissolved inorganic carbon (DIC), and dissolved organic carbon (DOC) export and investigated stream characteristics in the investigation catchment.

|  | Factors | d.f. | *p* | r$^2$ | Correlation |
|---|---|---|---|---|---|
| DC | Precipitation | 73 | 0.06 | 0.06 | + |
|  | Temperature | 73 | <0.01 ** | 0.12 | − |
|  | Sediment depth | 73 | 0.01 * | 0.10 | − |
|  | DC stock | 73 | <0.01 ** | 0.23 | + |
| DIC | Precipitation | 73 | 0.32 | 0.02 | + |
|  | Temperature | 73 | 0.03 * | 0.10 | − |
|  | Sediment depth | 73 | <0.01 ** | 0.17 | − |
|  | DIC stock | 73 | <0.01 ** | 0.24 | + |
| DOC | Precipitation | 73 | 0.12 | 0.04 | + |
|  | Temperature | 73 | 0.01 ** | 0.12 | − |
|  | Sediment depth | 73 | 0.01 ** | 0.12 | − |
|  | DOC stock | 73 | <0.01 ** | 0.26 | + |

* $p < 0.05$; ** $p < 0.01$; d.f.: degrees of freedom; +: positive; −: negative relationship.

## 4. Discussion

Our results indicated that the maximum values of the stocks of DC, DIC, and DOC were observed during the snowmelt season and had similar patterns, which was not consistent with our first hypothesis. Climatic factors (temperature and precipitation) and stream characteristics (sediment depth, discharge rate, and litter carbon input) drove the seasonal sink–source patterns for DC (DIC and DOC) in these streams. Furthermore, the results revealed that their exports showed trends similar to their stocks, which was also inconsistent with the second hypothesis. However, DC stock in the stream sources was higher than that at the ends of the streams in the forest–stream meta-ecosystem, which was in line with the third hypothesis, implying the subalpine forest streams may have important roles as carbon sources in subalpine forest regions.

### 4.1. Dynamics of DC (DIC and DOC) Stock in Forest Streams

In this study, the temporal variations in the DC, DIC, and DOC stocks were similar in all 15 streams, and the greatest peaks were observed during the snowmelt season (Figure 1), which was influenced by the precipitation and stream characteristics (discharge rate) along the streams (Table 2). The same situation also occurred in another study [34].

DIC stock in forest stream water is the bargain between the input and export. $HCO_3^-$ from the weathering carbonate rock and $CO_2$ dissolved from adjoined soil and air are the two main sources of DIC, which both are mainly regulated by climatic factors (temperature and precipitation) and stream characteristics [50,52]. In this study, the temperature fluctuated greatly during the snowmelt period, which, combined with the significant effects of freeze–thaw cycles, considerably enhanced the weathering of carbonate rocks [53–55]. In addition, the largest amount of precipitation occurred during this period, which increased stream discharge rates, leading to a greater release of DIC to the stream waters than during the other periods. Compared with the snowmelt period, the other periods, especially the growing season, showed higher mean temperatures and litter carbon inputs, which may have contributed more to the release of DIC than to the DIC yield. Higher temperatures accelerate $CO_2$ degassing from water, leading to a decrease in the DIC stock [56].

In addition, for the stocks of DC in the subalpine forest streams, because the leaching of soil from forest floors or the riparian zone by surface runoff is one of the main sources of DOC in streams [3], a greater amount of precipitation may cause more leaching, leading to more DOC being imported to the adjoining streams. Although many studies have shown that the decomposition of subalpine litter mainly occurs during the winter rather than during the growing seasons [57–59], most of this litter is covered by snow and ice and cannot be leached into the streams during the snow-covered season [60]. Therefore, the large fluctuations in temperature and the frequent freeze–thaw cycles induced litter fragmentation, allowing the litter to release more soluble substances and promoting the transportation

of DOC to streams with precipitation during the snowmelt season. Although there was more litter input to the streams during the growing season relative to the snowmelt season, most of this litter was not adequately decomposed via the interaction of water, soil, and sediments, causing a portion of these to be swept away as insoluble particles, while other portions became buried in the sediments.

All the situations mentioned above led to the maximum DC (DIC and DOC) stocks being observed during the snowmelt periods. The largest amount of precipitation occurred during the snowmelt period, which was also consistent with the seasonal dynamic patterns of the DC stocks at our sampling sites. Briefly, the DC stocks in streams are influenced by seasonal climatic changes and stream traits.

*4.2. Dynamics of DC Export from Streams*

The average export of DC in these subalpine forest streams ranged from 0.27–1.98 mg s$^{-1}$ for DC, 0.24–1.48 mg s$^{-1}$ for DIC, and 0.18–0.90 mg s$^{-1}$ for DOC, which was consistent with the results of other investigations [20,53]. Previous studies have shown a single peak during the snowmelt period, which was similar to the patterns of their stocks. The DIC and DOC exports from streams were mainly affected by their stocks, temperature, and sediment depth (Table 3). First, the DC stocks (both DIC and DOC) in stream water determine their exports from the streams. In other words, the more DC that was stored in the waters, the more DC that was exported from them, indicating that the DC stocks in the streams regulated their exports from the streams. Our results provided evidence that the DC stocks contribute most to their exports (Table 3). Second, although the higher temperatures during the growing periods relative to the snowmelt period may facilitate the degradation of litter and induce the release of more dissolved carbon to the stream water, the faster discharge rates caused by the increased rainfall during the growing and later growing seasons inhibits the accumulation of microbial residues, and most of the litter is leached without being decomposed [11,12,49], or it becomes buried in the sediments (more than 90% of the carbon; unpublished data). The higher temperature can also accelerate the degassing of DIC from water, and more DIC can be released as $CO_2$, thus resulting in a lower DC export rate than during the snowmelt season. The significantly negative relationship between temperature, sediment depth, and DC exports can also provide evidence.

Our results also showed that the exports of DIC and DOC accounted for 70% and 30% of the total DC stock, respectively. The ratios of DOC to DIC show that the form of carbon contributed by the subalpine forest streams was mainly DIC (Figure 5), which is consistent with a study by Butman et al. (2016) [53] but in opposition to results from Hobbie et al. One probable reason is that the streams studied by Hobbie et al. (1973) [61] were all located at the low elevation with higher annual precipitation, leading to less DIC in the water. Another reason might be the soil pH in the study was 4.3, containing fewer carbonates than in this research, while the amount of DIC was related to the amount of carbonates in the soil. The ratios of DOC to DIC in the export water were the lowest during the snowmelt period, which may be due to the heaviest precipitation that occurred in the snowmelt period, leading to more DIC export from stream water. They reached a maximum value during the later growing season, which may have mainly been caused by the sudden increase in DOC input during the later growing season [11]. Although the percentage of DIC and DOC exports from the forest streams were lower than those from soil respiration and plant respiration and the stream area was only 0.02% of the study area, a negative correlation between DC export and catchment area was also found in a study of DC export from 86 of Finland's main rivers and their catchments [40]. Therefore, the small headwater catchments in these subalpine forests might have a disproportionately large influence on terrestrial DC export. In addition, the importance of headwater catchments for DC export is further highlighted by the observation that most water enters stream networks via relatively small streams, which means the DC exports from the small subalpine forest streams are noteworthy parts of the forest ecosystem [20,62].

*4.3. Dynamics of the Ratios of DC in the Source Water to Export Water from Subalpine Forest Streams*

Our results found that the ratios of DC (DC, DIC, and DOC) stocks from source to end were mostly above one during the five different critical periods, except during the snowmelt season, which indicates that subalpine forest streams might serve as carbon sources. All these patterns appeared to be consistent with the variational periods, the maximum values being observed during the early growing period, and the minimum values being observed during the snowmelt periods (Figure 6), which may have mainly been caused by sudden precipitation during the snowmelt period, leading most of the DIC and DOC to be exported downstream [28,63].

We also found that the ratios showed different patterns in long and short streams (Figure 7). In streams longer than 30 m, the total DC and DIC stocks were in the order source > estuary > middle, while for streams of a length less than 30 m, the stocks of total DC and DIC were in the order estuary > source > middle (Table 4). Their DOC stocks showed that the estuary > source and the middle of the stream were not consistent. Our field investigation found one possible reason: the headwaters of longer streams were often affected by vegetation canopy cover, slow discharge rates, more plant residues, and more dissolved and accumulated carbon. However, the short streams, which were close to the main river channel and had higher discharge velocities, lacked plant residues and biological activity. While the DC in the streams mainly originates from scouring and bleaching processes, the DC stock in the streams into estuaries is relatively higher [20,64]. The possible reason for a downstream decrease in DOC included changing hydrological pathways through soils and increasing proportions of postglacial sediments deposited in the lower-lying parts of a catchment, which reduced export because of DOC adsorption to mineral surfaces. Thus, investigations on the DC dynamics for forest streams can provide baseline data for in-depth research on the internal relationships between individual aquatic and terrestrial ecosystems within a forest–stream ecosystem.

**Table 4.** Export rates of dissolved carbon (DC), dissolved inorganic carbon (DIC), and dissolved organic carbon (DOC) of the 15 streams in the subalpine forest stream.

| Stream | Stream Length (m) | DC (mg s$^{-1}$) | DIC (mg s$^{-1}$) | DOC (mg s$^{-1}$) |
|--------|-------------------|------------------|-------------------|-------------------|
| A | 37.6 | $1.44 \pm 0.20$ | $0.97 \pm 0.16$ | $0.72 \pm 0.08$ |
| B | 20.8 | $1.24 \pm 0.16$ | $0.85 \pm 0.09$ | $0.49 \pm 0.06$ |
| C | 62.3 | $0.64 \pm 0.10$ | $0.36 \pm 0.06$ | $0.27 \pm 0.03$ |
| D | 225.6 | $0.87 \pm 0.07$ | $0.52 \pm 0.10$ | $0.35 \pm 0.04$ |
| E | 108.0 | $0.13 \pm 0.02$ | $0.06 \pm 0.08$ | $0.05 \pm 0.06$ |
| F | 186.0 | $1.04 \pm 0.14$ | $1.55 \pm 0.14$ | $0.92 \pm 0.10$ |
| G | 92.4 | $1.09 \pm 0.14$ | $0.64 \pm 0.08$ | $0.46 \pm 0.06$ |
| H | 132. | $0.81 \pm 0.07$ | $0.52 \pm 0.06$ | $0.28 \pm 0.03$ |
| I | 66.0 | $1.29 \pm 0.14$ | $1.00 \pm 0.12$ | $0.73 \pm 0.08$ |
| J | 18.0 | $1.39 \pm 0.15$ | $0.77 \pm 0.09$ | $0.62 \pm 0.07$ |
| K | 36.0 | $1.78 \pm 0.19$ | $0.99 \pm 0.11$ | $0.80 \pm 0.09$ |
| L | 11.3 | $0.95 \pm 0.08$ | $0.61 \pm 0.07$ | $0.37 \pm 0.05$ |
| M | 27.6 | $0.35 \pm 0.04$ | $0.21 \pm 0.03$ | $0.16 \pm 0.02$ |
| N | 32.0 | $1.52 \pm 0.25$ | $1.15 \pm 0.13$ | $0.74 \pm 0.08$ |
| O | 86.0 | $0.51 \pm 0.05$ | $0.30 \pm 0.05$ | $0.21 \pm 0.03$ |

Each export rate in the table is the mean and standard deviation of 13 sampling times for each forest stream. A–O are the sampled streams in the study.

## 5. Conclusions

The total stock and the export of DC (DIC, DOC) in the streams of the subalpine forest represented the same patterns during different periods, as influenced by seasonal changes in temperature, precipitation, and stream characteristics. Seasonal precipitation patterns and stream discharge rates dominated the total stock of DC in the streams. Meanwhile, the seasonal temperature, depth of sediment, and DC stock mainly regulated the exports of DC from the streams. The DC stock at the source of the streams was higher than at the

ends of the streams in this ecosystem, implying that subalpine forest streams may act as carbon sources and play crucial roles in linking the carbon cycles between forests and their adjoining streams. The DOC: DIC ratios in water might provide a potential parameter for developing both the modelling and databases needed to understand fully the role of inland waters in the carbon cycle. Future investigations on carbon biogeochemicals within the forest meta-ecosystems should be conducted, especially exploring the inner relationships between C biogeochemical cycle in these meta-ecosystems and forest properties, topographical features, and catchment traits under different climate change scenarios.

**Author Contributions:** Conceptualization, W.Y., and J.H.; methodology, J.H., and W.Y.; software, J.H. and F.L.; validation, J.H. and W.Y.; formal analysis, J.H.; investigation, J.H., F.L., Z.W., X.L. and R.C.; resources, W.Y.; data curation, W.Y. and J.H.; writing—original draft preparation, J.H.; writing—review and editing, W.Y. and J.H.; visualization, J.H.; supervision, W.Y.; project administration, W.Y.; funding acquisition, W.Y. and F.L. All authors have read and agreed to the published version of the manuscript.

**Funding:** This study was supported by the National Natural Science Foundation of China (32071554, 32001139).

**Institutional Review Board Statement:** Not applicable.

**Informed Consent Statement:** Not applicable.

**Data Availability Statement:** All data generated or analyzed during this study are included in this published article. No new data were created or analyzed in this study. Data sharing does not apply to this article.

**Acknowledgments:** We thank all contributors for their kind assistance in the development of this manuscript.

**Conflicts of Interest:** The authors declare no conflict of interest.

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
