# Peer review of "Dynamics of Dissolved Carbon in Subalpine Forest Streams"

_forests, doi:10.3390/f13050795_

Round 1

Reviewer 1 Report

The authors have provided a well written manuscript. The topic of the research is not new, but the research has been conducted with the required care. The presentation of the results is clear and comprehensive.

Overall, the manuscript is more a stock-taking on carbon fluxes in one particular ecosystem. As such it is a useful reference.

The authors did not provide informatiion on the expected relevance of their findings. As presented, the text can be useful for colleagues who want to obtain an overview on the subject.

Author Response

Dear Reviewer:

   Thank you for your comments concerning our manuscript entitle “Dynamics of dissolved carbon in subalpine forest streams”. Those comments are all valuable and very helpful for revising and improving our paper, as well as the important guiding significance to our research. We have studied the comments carefully and have made corrections which we hope to meet with approval. Revised portions are marked in red on the paper. Please see the attachment.

Reviewer 2 Report

The present manuscript could be seen as a very preliminary study on forest streams since very few studies investigated such streams.
The manuscript need to be throughly revised and I suggest an English editing becuse many sentences are hard to understand.

The title should be changed because you did not investigate the dynamics but just the quantity of DC in streams

Ls 30-32: Remove this part and delete Figure 1. The introduction section could start as follow: "Forest streams are conduits....."

L 33: are you sure about this statement? the stream water just transports C. The soil leaching process remove organic carbon from the terrestrial ecosystem

L 35: forests and riparian vegetation could be the major sources of organic carbon, not just carbon

Ls 37-38: major form of input for what? Which kind of C? organic? inorganic?

Ls 46-50: you could just write "DC includes dissolved organic and inorganic C (DOC and DIC, respectively) which influence the surface water chemistry, e.g., the metal ......."

L 50: it is true for all kind of ecosystems, not just for forest and riparian ones

L 54: maybe you intend vegetation stages?

Ls 54-55: characteristics of what?

L 55: DC storage...where?

Ls 55-59: this part should be shortened

Ls 60-62: I think that it is true for all kind of streams

L 71: what do you mean for "critical periods"?

Ls 74-75: it is true for all rivers

Ls 81-102: in my opinion this part can be deleted because most of the factors that you report are not taken in account in your investigation (e.g., landform, primary production, etc.), and there are some repetions

Ls 103-117: this part should be shortened and the aims must be better clarified

L 125: information about geology and soil types of the study site are missing

Table 1: the column "variable" can be removed. What does the 0-5 cm soil depth interval include? Just the organic horizon or also the uppermost mineral horizon? Having a look to C, N and P concentrations, you considered the organic horizon. Does it include the fresh litter? Maybe it is usefull to add the mean concentration of carbonates of soil

Ls 175-183: what are the units of these parameters? please, add them

L 176: it is not clear what "V" is

Ls 186-188: how did you test the ANOVA assumptions? Please, report

L 189: the sampling period should be treated as random effect

Ls 191-196: this part should be clarified, it is quite confusing

Figure captions: what is the meaning of the error bars?

Ls 219-227: Remove this part

Ls 248-255: remove this part

L 270: these calculations are not reported in materials and methods

Figure 7: improve the quality of this figure

Figure 8 is similar to Figure 7

Ls 306-309: which ones were negatively correlated and which ones were positively correlated?

Ls 315-325: remove this part

Ls 323-325: what do you intend for self-purification? 

Ls 331-332: not clear statement. Reword

L 335: in this manuscript, figures showing the seasonal changes of temperatures and precipitations are missing

L 346: replace erosion with leaching

Ls 348-349: are you sure? The lower temperatures decrease the litter decomposition

Ls 353-357: remove this part

L 356: if they are not soluble, they cannot be leached

Ls 369-372: unclear, reword

L 377: do you mean microbial necromass? If so, you did not analysed it

L 378: not clear. You did not measure the quality of the organic matter. Probably you are confusing the leaching with the erosion process

Ls 378-379: you did not investigate the sediments, please remove this statement

Ls 384-390: very long sentence. Further, the amount of DIC is related to the amount of carbonates in soil. Probably Zhou and Hobbie studied two different soil types

Ls 407-410: I note that the values for DC and DIC are very close to 1. It means that their concentrations do not change from the source to the end of the investigated streams. Some changes occurred just for DOC, where in some cases there is an increase of DOC and in others a decrease. Anyway, the term "purification" is not appropriate

Ls 415-426: You did not performe any statistical analysis between streams longer than 30 m and streams long lesser than 30 m.. Remove this part

Author Response

(The authors gave the same response as above.)

Reviewer 3 Report

This manuscript presents an interesting study, which measured and analyzed the dissolved carbon stock and export in subalpine forest streams. It does not have any obvious scientific questions, only requires clarifying some concepts and adding more information. In addition, there are some grammar and tense errors, please double-check your writing.

  1. Line 9: Delete ‘and carbon cycling models’. A lot of models include this carbon flux process. van Hoek et al. (2021) has a brief summary of these models. Just delete this statement.
  2. Line 10: Delete ‘the carbon’.
  3. Line 16: change ‘;’ to ‘,’. Please double-check this issue throughout your manuscript.
  4. Abstract: Please include your finding (the DIC:DOC ratio) in the abstract, this is a significant contribution to riverine OC studies. It can be used in model development.
  5. Line 26: Please use alphabetical order to arrange these keywords.
  6. Line 30-41: Delete figure 1. This illustration figure could not deliver any useful information. If you want to keep it, please look at figure 1 in Cole et al. (2007) and revise your figure 1. DC includes DIC and DOC, right? Please clarify it.
  7. Line 50: revise the citation format (Hruška et al., 2003).
  8. Line 50-52: I am not fully understanding this sentence. import or export? Please revise it.
  9. Line 60: delete ‘From a theoretical perspective,’
  10. Line 81-82: Please clearly address the terrestrial-aquatic DIC and DOC fluxes. They initially flux from soils to inland waters. In aquatic ecosystems, partial OC is released into the atmosphere through outgassing or buried in the sediment, and the remainder is eventually delivered from the landscape. See Cole et al. (2007) (figure 1) and Wei et al. (2021a).
  11. Line 85-86: Delete ‘other biotic and abiotic factors [31-33]’. Please add ‘nitrogen and sulfur deposition’. Nitrogen and sulfur deposition can accelerate the soil DOC leaching by increasing the soil DOC production and controlling the soil sorption ability. See Wei et al. (2021b).
  12. Line 103: Delete ‘Since they are frequently affected by geological disasters, earthquakes, landslides, debris flows and collapses [45],’ and rewrite the whole sentence. Geological disasters are not frequent. You may introduce your scientific question by discussing disturbances (fire, heavy precipitation).
  13. Line 112 – 114: Delete this sentence. It is better to highlight the contribution of your study to science (model development, carbon budget estimation) other than ‘lacking’.
  14. Table 1: Move ‘MAP: mean annual precipitation; MAT: mean annual temperature; C, N, P values reported were concentrations in the surface soil (0-5 cm).’ to the table caption.
  15. Line 126: Can you add a map of your study area? Please include the land cover information and these rivers.
  16. Line 178: flow – discharge, please revise this issue in this manuscript.
  17. Line 228: Please use ‘export’ to replace ‘output’.
  18. Figures 7 and 8 are too vague, please update them.
  19. Tables 2 and 3, can you plot these data? I understand that each figure may have 12 panels, but it should be important to see your data. Is your data public? Can we access them?
  20. The DIC/DOC ratio is rarely reported in existing studies. It should be interesting if you can compare your results with existing studies and highlight them in the conclusion section. Maybe you can get useful information from Butman et al. (2016).

Butman, D. et al., 2016. Aquatic carbon cycling in the conterminous United States and implications for terrestrial carbon accounting. Proceedings of the National Academy of Sciences, 113(1): 58-63.

Cole, J.J. et al., 2007. Plumbing the global carbon cycle: integrating inland waters into the terrestrial carbon budget. Ecosystems, 10(1): 172-185.

van Hoek, W.J. et al., 2021. Exploring Spatially Explicit Changes in Carbon Budgets of Global River Basins during the 20th Century. Environmental science & technology, 55(24): 16757-16769.

Wei, X. et al., 2021a. Climate and atmospheric deposition drive the inter-annual variability and long-term trend of dissolved organic carbon flux in the conterminous United States. Science of The Total Environment, 771: 145448.

Wei, X. et al., 2021b. Identifying key environmental factors explaining temporal patterns of DOC export from watersheds in the conterminous United States. Journal of Geophysical Research: Biogeosciences, 126(5): e2020JG005813.

Author Response

Dear Reviewer:

   Thank you for your comments concerning our manuscript entitle “Dynamics of dissolved carbon in subalpine forest streams”. Those comments are all valuable and very helpful for revising and improving our paper, as well as the important guiding significance to our research. We have studied the comments carefully and have made a correction which we hope to meet with approval. Revised portions are marked in red on the paper. Please see the attachment。

Round 2

Reviewer 2 Report

I appreciate the effort of the authors to improve the manuscript, but a further check is needed

L 129: within the cited article, soil was not classified. Once classified, you should cite the classification system used

Ls 174-190. I suggest you to use different abbreviations for stock and for stream and catchment areas. Both stock and areas have "S" as abbreviation. There are some errors for the units. for example, according to the equation, the stock unit should be mg L-1 m. For K (eq. 2) the unit should be mg m3 L-1 s-1.  Also F unit is wrong.
In my opinion, the calculation of stock for streams does not make sense because the water does not stand inside the stream. Anyway, in my opinion the stock should be expressed as mg m-1

L 198: which post-hoc test did you use to compare the means? please report

L 199: why did you use the linear mixed-effect models instead of Spearman or Pearson correlation test for the correlation?

L 209: at line 175 you reported that for stock the unit was mg L-1

L 242: check the unit also for the export rate

Table 2: The r2 values seem to be too low for a significant correlation in particular for those with r2<0.10, please check. This could cause changes for the discussion section

Table 3: please check the correlations, the r2 values seem to be too low to give significant correlations

L 304: please, add the full name of the abbreciations "Export rates of dissolved carbon (DC), dissolved inorganic carbon (DIC),......"

L 341: ok, they are means and standard error or standard deviation?

Author Response

Dear Reviewer:

    Thank you again for your comments concerning our manuscript entitled “Dynamics of dissolved carbon in subalpine forest streams”. Those comments are all valuable and very helpful for revising and improving our paper, as well as the important guiding significance to our research. We have studied the comments carefully and have made corrections which we hope to meet with approval. Revised portions are marked in red on the paper. Please see the attachment.

Reviewer 3 Report

The manuscript has been well revised. It is ready to be published.

Author Response

Dear Reviewer:

     Thank you again for your comments concerning our manuscript entitled “Dynamics of dissolved carbon in subalpine forest streams”.